# Physico-Chemistry of Dinitrosyl Iron Complexes as a Determinant of Their Biological Activity [note 1]

**DOI:** 10.3390/ijms221910356

**Published:** 2021-09-26

**Authors:** Anatoly F. Vanin

**Affiliations:** N.N. Semenov Federal Research Center for Chemical Physics, Russian Academy of Sciences Moscow, 119991 Moscow, Russia; vanin@polymer.chph.ras.ru

**Keywords:** nitric oxide, nitrosonium cation, dinitrosyl iron complexes, thiol-containing ligands, S-nitrosothiols

## Abstract

In this article we minutely discuss the so-called “oxidative” mechanism of mononuclear form of dinitrosyl iron complexes (M-DNICs) formations proposed by the author. M-DNICs are proposed to be formed from their building material—neutral NO molecules, Fe^2+^ ions and anionic non-thiol (L^−^) and thiol (RS^−^) ligands based on the disproportionation reaction of NO molecules binding with divalent ion irons in pairs. Then a protonated form of nitroxyl anion (NO^−^) appearing in the reaction is released from this group and a neutral NO molecule is included instead. As a result, M-DNICs are produced. Their resonance structure is described as [(L^−^)_2_Fe^2+^(NO)(NO^+^)], in which nitrosyl ligands are represented by NO molecules and nitrosonium cations in equal proportions. Binding of hydroxyl ions with the latter causes conversion of these cations into nitrite anions at neutral pH values and therefore transformation of DNICs into the corresponding high-spin mononitrosyl iron complexes (MNICs) with the resonance structure described as [(L^−^)_2_Fe^2+^(NO)]. In case of replacing L^−^ by thiol-containing ligands, which are characterized by high π-donor activity, electron density transferred from sulfur atoms to iron-dinitrosyl groups neutralizes the positive charge on nitrosonium cations, which prevents their hydrolysis, ensuring relatively a high stability of the corresponding M-DNICs with the resonance structure [(RS^−^)_2_Fe^2+^ (NO, NO^+^)]. Therefore, M-DNICs with thiol-containing ligands, as well as their binuclear analogs (B-DNICs, respective resonance structure [(RS^−^)_2_Fe^2+^_2_ (NO, NO^+^)_2_]), can serve donors of both NO and NO^+^. Experiments with solutions of B-DNICs with glutathione or N-acetyl-L-cysteine (B-DNIC-GSH or B-DNIC-NAC) showed that these complexes release both NO and NO^+^ in case of decomposition in the presence of acid or after oxidation of thiol-containing ligands in them. The level of released NO was measured via optical absorption intensity of NO in the gaseous phase, while the number of released nitrosonium cations was determined based on their inclusion in S-nitrosothiols or their conversion into nitrite anions. Biomedical research showed the ability of DNICs with thiol-containing ligands to be donors of NO and NO^+^ and produce various biological effects on living organisms. At the same time, NO molecules released from DNICs usually have a positive and regulatory effect on organisms, while nitrosonium cations have a negative and cytotoxic effect.

## 1. Introduction

We have collected a lot of experimental material showing that researchers have a new class of chemical compounds at their disposal—dinitrosyl iron complexes (DNICs) with thiol-containing ligands. These compounds can serve the basis for designing medicines with a wide range therapeutic action relying on the fact that nitrogen monoxide (or nitric oxide, NO) is one of the main components of DNICs [1,2,3,4,5,6,7,8,9,10,11,12,13,14,15,16,17,18,19,20,21,22]. This is a remarkably simple compound containing two elements that form part and parcel of Earth’s atmosphere: nitrogen and oxygen. The compound is continuously produced in all living organisms through fermentation and is one of versatile regulators of many physiological and biochemical processes [23,24,25]. NO content in DNICs with thiol-containing ligands as well as the ability of these complexes to release this extremely bioactive compound attracts attention of an increasing number of researchers in various fields of biology and medicine [26,27,28,29,30,31,32,33,34,35].

Including nitric oxide in DNICs with thiol-containing ligands ensures not only the stabilization of NO and its transfer to biological targets, but also the conversion of NO molecules into nitrosonium cations (NO^+^), due to which DNICs with thiol-containing ligands are capable of initiating S-nitrosation of low- and high-molecular-weight thiol-containing compounds, i.e., processes orchestrated by a system of endogenous nitric oxide [36,37,38,39,40].

DNICs with thiol-containing ligands are easily chemically synthesized. The most interesting fact is that they are also produced within living organisms when there is nitric oxide. Moreover, we have reasons to consider these complexes as a “working form” of NO that ensures various types of biological activity of this agent [41].

Therefore, my review aims to scrutinize how physical and chemical properties of the original “building material” of these complexes—divalent iron ions, NO molecules and thiol-containing compounds—determine the formation mechanism of DNICs with thiol-containing ligands. The latter determines various aspects of biological activity of these complexes, and mainly their ability to have an impact as NO and NO^+^ donors on living organisms.

## 2. History of Discovery and Identification of DNICs with Thiol-Containing Ligands in Living Organisms

DNICs with thiol-containing ligands were discovered in the 1960s, first in baker’s yeast cells, and then in animal tissues. That was closely connected with interest in the role of free radicals in biological systems which was widely expressed at that time. That interest was initiated by the fact that the famous Russian scientist and Nobel Prize winner N. Semyonov discovered extremely chemically active compounds—free radicals—initiating and supporting chain reactions.

It was attractive to assume that high efficiency of metabolic pathways in living organisms occurring at rather low temperatures (≤40 °C) was due to free radicals formed in those organisms. This assumption was verified using the recently developed method of detecting free radicals with paramagnetic nature based on electron paramagnetic resonance spectroscopy (EPR spectroscopy) and showed that substrates of many biochemical redox reactions are converted into free radicals. This is detected based on relatively narrow symmetrical EPR signals with the width of 12–15 Oersteds (1.2–1.5 mT) and the g-factor of the center of signals which is 2.0 that emerge in the EPR spectrum of biological objects [42]. Judging by these characteristics, free radicals which produce these signals were organic compounds including such “light” elements as hydrogen, nitrogen, oxygen, and phosphorus.

An absolutely different type of EPR signals that came from free radicals, as we initially assumed for the some time, was detected by me in collaboration with Robert Nalbandyan in baker’s yeast (*Saccharomyces cerevisiae*) cells (XIY), when we studied those cells via the EPR method [43,44,45]. In the preparations obtained by drying yeast at 40–45 °C or their lyophilization, apart from the above-mentioned EPR signal of organic free radicals at g = 2.0, we registered the EPR signal of a new type with the main peak at g = 2.04 (Figure 1A). Subsequent research showed that that peak was a component of the EPR signal with g_⊥_ = 2.04, g_‖_= 2.014, g_aver._ = 2.03, titled the 2.03 signal according to the value of g_aver._ (Figure 1B).

The form of this signal in dried yeast preparations did not change when the recording temperature was changed, ranging between liquid nitrogen temperature (−273 °C) to 90–100 °C (Figure 1C), and under the conditions of microwave power saturation of the 2.03 signal (data not provided). These facts allowed us to draw the conclusion that the components of the 2.03 signal at g_⊥_ and g_‖_ were really those of the same signal. Of note, the amplitude of the 2.03 signal in the mentioned temperature range was in accordance with Curie’s law for paramagnetic compounds.

When dried yeast preparations were kept at 90–100 °C, the 2.03 signal irreversibly disappeared [44]. At the same time, the signal form, namely the proportion of component amplitude at g_⊥_ and g_‖_ remained unchanged. When moist yeast was heated in their growth medium the 2.03 signal disappeared in them at 50–60 °C [45,46]. This provided the ground to conclude that the centers responsible for that signal consisted of proteins. That conclusion was confirmed when we managed to register the 2.03 signal at room temperature in the water-containing yeast preparation (Figure 1D) [45]. The fact that there remained the anisotropic (asymmetrical) 2.03 signal with the half-width of ~4.0 mT clearly showed that the centers responsible for that signal were connected with a macromolecular (protein) matrix. Low mobility of that matrix at room temperature obviously could not provide for an averaging of g-factor anisotropy, as it would be if the centers responsible for the 2.03 signal were of low molecular weight.

Considerable deviation of the g-factor of the 2.03 signal from purely spin values (2.0024) showed that an unpaired electron was localized in the centers responsible for that signal at relatively “heavy” atoms in those centers. Such a deviation was detected for EPR signals, reported at that time (the 1960s), with unpaired electrons localized primarily on sulfur atoms [47,48,49]. As our experiments with treatment of yeast that gave the 2.03 signal with heavy metal ions bound with thiol sulfur atoms—Ag^+^, Hg^2+^ or Cu^2+^ salts—caused the 2.03 signal to completely disappear, it was assumed that paramagnetic centers responsible for that signal were thiyl (“sulfur”) free radicals [45,46]. However, there was one sharp difference between EPR signals of those radicals and the 2.03 signal—one of the three values of the g-factor tensor for the EPR signal of “sulfur” radicals was close to a purely spin value, while for the 2.03 signal, the minimum value of that tensor (2.014) was considerably different from g = 2.0024. That important fact caused doubts regarding the assumption about the “sulfur” nature of the centers responsible for the 2.03 signal.

We made the next step in identifying those centers after Azhipa, Kaiushin and Nikishkin had registered the 2.03 signal in the lyophilized preparation of liver mitochondria of rats administered with sodium nitrite (Figure 2B) [50]. After comparison of this signal with the EPR signal of hemoglobin nitrosyl complex registered in those animals’ blood (Figure 2A) they concluded that the centers responsible for the 2.03 signal in mitochondria were also nitrosyl complexes of heme-containing mitochondrial proteins—terminal cytochromes.

The main argument for such identification was that the average g-factor value of the 2.03 signal and the g-factor of the middle of the triplet hyperfine structure (triplet HFS) in the nitrosyl hemoglobin complex spectrum coincided (Figure 2). However, three years later, the same authors reported that the g-factor of that middle provided by them was a misprint. It proved to be 2.01 [51]. Therefore, identification of the nature of the centers responsible for the 2.03 signal by those authors had no more solid ground.

Identifying the 2.03 signal in rat mitochondria treated with sodium nitrite allowed us to immediately assume that NO participates in the formation of the centers responsible for that signal. The appearance of the latter in rat organism was proved by registration of the EPR signal of nitrosyl hemoglobin complexes in animals’ blood; that signal had previously been detected in [48]. Certainly, there were no reasons to assume that the centers responsible for the 2.03 signal and the signal of nitrosyl hemoglobin complexes as representatives of heme-containing proteins, that were sharply different in terms of the form and values of the g-factors, were of the same nature.

That difference made me switch to other iron-containing mitochondrial proteins—the so-called iron-sulfur proteins—whose reaction with NO could give rise to nitrosyl iron complexes that give the 2.03 signal. That assumption was supported by the fact that active centers of those proteins included protein thiol-containing ligands, apart from iron and inorganic sulfur [52]. It would be impossible to verify that assumption at that time, because we could hardly obtain iron-sulfur proteins from animal tissues in a sufficiently pure form. Fortunately, at that time, in 1965, there was article by the American chemists C. McDonald et al. [53], where EPR signals of DNICs with various low-molecular-weight anion (including thiol-containing) ligands registered at room temperature were provided (Figure 3A). Certainly, due to high mobility of low-molecular-weight DNICs, sufficient for averaging the possible anisotropy of the g-factor and triplet HFS, those signals were narrow symmetric singlets with the g-factor of the signal center (actually with g_aver_) equal to 2.03).

The latter, as well as the apparent contribution of thiol-containing ligands in low-molecular-weight DNICs, led me to verify whether the latter were an exact copy of the centers responsible for the 2.03 signal in yeast. As our research revealed, those centers most probably consisted of protein, which could cause and indeed caused the fact that that the 2.03 signal in yeast registered at room temperature had a relatively big width and anisotropic form stemming from the anisotropy of g-factor tensor (Figure 1D).

Therefore, to verify that, it was necessary to register the EPR signal of low-molecular-weight DNICs with thiol-containing ligands, for instance, with cysteine, at a low temperature (such as liquid nitrogen temperature) to block any mobility of those complexes and to find out what their EPR signal form would be in that case. It was identified that in those conditions the EPR signal form was the same as that of the 2.03 signal (Figure 4d) [54]. When I performed the same with DNICs with non-thiol ligands, the form of registered signals differed from the 2.03 signal (Figure 4a–c).

That difference for DNICs with water or phosphate was determined by the low (rhombic) symmetry of those complexes, whose EPR signal was described, as distinct from the 2.03 signal, by three different values of the g-factor tensor—g_1_ = 2.05, g_2_ = 2.03 and g_3_ = 2.014, rather than two (Figure 4b′,c′) Another representative of DNICs with non-thiol ligands, DNICs with hydroxyl anions, were characterized by the EPR signal with two different g-factor tensor values— g_⊥_ = 2.014, g_‖_ = 2.04; however, the ratio between its values, g_⊥_ = 2.014, g_‖_ = 2.04 (g_‖_ > g_⊥_) (Figure 4a′), was contrary to the ratio of those values for the 2.03 signal and the signal for DNICs with cysteine g_⊥_ = 2.04, g_‖_ = 2.014 (g_⊥_ > g_‖_) (Figure 4d′). The fact that there were only two different g-factor tensor values for EPR signals of DNICs with hydroxyl and cysteine was the evidence of axial symmetry of those complexes.

Initially I assumed that the coincidence of the 2.03 signal characteristics and the characteristics of the EPR signal of frozen solutions of DNICs with thiol-containing ligands did not mean that the centers responsible for the 2.03 signal really were DNICs with thiol-containing ligands. It was assumed that, instead of NO, those centers included some nitrogen-containing compounds whose nature was close to NO. However, it quickly became clear after experiments with treating yeast and isolated animal tissues with gaseous NO or nitrite and nitrate, or after experiments with including the latter in the drinking of mice, due to which the 2.03 signal was registered in yeast and animal tissues, that the centers responsible for that signal were really DNICs with thiol-containing ligands [55]. At the same time, the latter can be both low-molecular-weight and protein thiols.

In the 1967 publication we showed that DNICs with thiol-containing ligands could appear in animal tissues not only in vitro, i.e., after treatment of isolated tissues with gaseous NO or nitrite, but also by themselves in vivo (Figure 5C) [56] as well as in yeast cells (Figure 5A) [46]. It was assumed that in that case DNIC formation was initiated by introduction of nitrite as NO donor into the animal organism [56].

It should be noted that we were not the only researchers who first observed the 2.03 signal first in yeast cells (1963–1966), and then in animal tissues (1967–1969). In 1964 English researchers Mallard and Kent paid attention to the appearance of a weak EPR signal at g = 2.03 in the EPR spectrum of chemically induced rat hepatoma (Figure 5B, spectrum b), yet did not mention it any more [57]. As distinct from them, American researchers Commoner and his colleagues, who registered a more noticeable component of the 2.03 signal at g = 2.035 during similar experiments with rats in 1965 (Figure 5D) (i.e., only part of the 2.03 signal) responsible for that signal [58], started studying the nature of the centers more actively.

For the first time Commoner and colleagues considered those centers to be free (organic) radicals [58]. It is interesting that the 2.03 signal, or more precisely, its component at g = 2.035, registered in the work by Commoner and colleagues, appeared a few days after introducing hepatocarcinogenic agents into rat organism, and then it disappeared within a week. After that, malignization of liver tissue started [58]. This allowed me to assume that the 2.03 signal could be considered as an indicator of tissue malignization. However, we detected the intensive 2.03 signal in rabbit liver (1967) without any signs of malignization implying that the assumption was incorrect [56].

In the work of 1968 Commoner and his colleagues in fact repeated our experiments demonstrating a complete similarity of the 2.03 signal in biological objects and frozen solutions of DNICs with thiol-containing ligands [59]. Finally, in subsequent research (the publication of 1970) [60], after establishing the correlation between registering the 2.03 signal in rat liver and introduction of nitrate in their organisms with drinking water, the authors agreed with our conclusion that the centers responsible for the 2.03 signal in living organisms are represented by DNICs with thiol-containing ligands [61].

Based on our finding and the results obtained by Commoner and his colleagues suggesting that the 2.03 signal was hypothetically generated by DNICs with thiol-containing ligands in the work of 1990 Lancaster and Hibbs proposed that these DNICs were generated by activated rat macrophages in the presence of endogenous NO formed from L-arginine under the catalytic effect of inducible NO-synthase (iNOS). When activated macrophages were treated with iNOS inhibitor N-methyl-L-arginine (NMMA), the attenuation of the 2.03 signal was observed (Figure 5, right panel) [62].

In connection with that, it is interesting to quote from recent work of Lancaster [63], where discovery and identification of DNICs in living organisms are presented in the following way: “In 1965 independently Commoner and Vanin described a new EPR signal with a principal g-value of 2.03 in precancerous rat liver (Commoner) and in yeast (Vanin). The molecular origin of these species was identified by McDonald et al. (1965) as an NO complex with iron and several type of ligand. Soon afterward, Azhipa, Kaiushin and Nikishkin concluded that this the origin of the signal in yeast and liver”. Here we see that Dr. Lancaster completely distorted the real history behind identifying the centers responsible for the 2.03 signal, which I primarily conducted in the 1960s.

## 3. Mechanisms of Formation of DNICs with Thiol-Containing Ligands in Living Organisms

Analysis of the isotropic 13-component HFS shown in Figure 3A and characteristic of the EPR signal of DNICs with cysteine first registered at room temperature [53] already showed that that the complex included two nitrosyl ligands and two cysteine molecules. The 13-component HFS emerges as a result of interaction between an unpaired electron with two nitrogen (^14^N) nuclei (with the nuclear spin I = 1) of two nitrosyl ligands and four methylene group protons (with I = 1/2) in two cysteine molecules included in DNICs (Figure 6d). When we substituted ^14^NO in DNICs with cysteine by ^15^NO with I = ½ instead of the 13-component HFS, the nine-component HFS was observed, and we deciphered it in Figure 6f.

As far as the number of iron ions included in DNICs with cysteine is concerned, the appearance of the relevant doublet hyperfine splitting of the EPR signal of those complexes when including ^57^Fe (I = 1/2) in them (Figure 6h) certainly indicated that those DNICs were mononuclear (with one iron atom).

Therefore, the analysis of the HFS of the EPR signal of DNICs with thiol-containing ligands already showed that those complexes were mononuclear iron complexes that included two thiol-containing ligands and two nitrosyl ligands each. That conclusion was completely corroborated by the results of X-ray analysis of crystalline samples of those complexes [64,65,66]. As regards the spin state of those complexes, registration of their EPR signals at room temperature clearly indicated their low-spin nature with S = 1/2 [39,53,67].

As shown in the work by McDonald et al. [53], DNICs are formed with divalent iron, and the main question, which immediately arises when the mechanism of those complexes is considered, is the following. How does the binding of two free-radical NO molecules, each having one unpaired electron on the highest molecular orbital (MO), with a Fe^2+^ ion with six electrons on its 3d orbitals, cause the appearance on MO of a produced Fe(NO)_2_ group of an odd number of electrons characteristic of the paramagnetic state of that group (instead of their even number equal to eight, obtained by adding two electrons of nitrosyl ligands to six iron electrons characteristic of diamagnetic or high-spin state of that group)? The only possible answer to this question is: the odd number of electrons on MO of the Fe(NO)_2_ group can be the result of one-electron oxidation or reduction of that group. Due to that, the total number of electrons on MO of the Fe(NO)_2_ group will be seven or nine respectively. As represented by Enemark-Feltham, this corresponds to the formulae of iron-dinitrosyl groups - {Fe(NO)_2_}^7^ or {Fe(NO)_2_}^9^ [68], formally corresponding to d^7^ or d^9^ electron configurations of iron in those groups respectively.

The processes of one-electron oxidation or reduction of the Fe(NO)_2_ group are the basis for the currently proposed mechanisms for forming DNICs with thiol-containing ligands in both living organisms and purely chemical systems. The first mechanism is oxidation causing {Fe(NO)_2_}^7^ representation of the iron-dinitrosyl group (or formally d^7^ electron configuration in that group), and it was already proposed by our group in late 1990s [69]. The second mechanism of reduction which determines formally d^9^ electron configurations of iron in DNICs or {Fe(NO)_2_}^9^ representation of the iron-dinitrosyl group in those complexes, and it was proposed not long ago in the works of P. Ford’s group [33,34,70,71].

We proposed the oxidative mechanism for DNIC formation when my colleague Andrew Thomson from the UK attracted my attention to an article by American chemists published in the 1980s, where they found accumulation of N_2_O during formation of DNICs with non-thiol ligands [72]. The authors explained that the phenomenon as related to reduction of NO in that system to N_2_O due to Fe^2+^ ions. However, that did not correspond to Mössbauer spectra of DNICs with cysteine measured by us by that time [73]. Those spectra demonstrated Mössbauer absorption characteristics of DNICs only, without the absorption caused by iron ions not included in DNICs. Therefore, we assumed [69] that N_2_O appearing during DNICs formation could be related to the disproportionation reaction of two NO molecules, bound with Fe^2+^ ions in pairs, i.e., the reaction of mutual one-electron oxidation and reduction of those molecules causing their conversion into a nitrosonium cation (NO^+^) and nitroxyl anion (NO^-^) in accordance with Equation (1):2NO ⇆ NO^+^ + NO^−^(1)

NO^+^ and NO^−^ appearing during this reaction are hydrolyzed during contact with water, forming NO_2_ and N_2_O respectively in accordance with the reaction presented by Equation (2):3NO ⇆ NO_2_ + N_2_O(2)

Such transformation of NO^+^ and NO^−^ into NO_2_ and N_2_O, respectively, is illustrated by the following algebraic transformation of Equation (1). After multiplying the latter by two and adding four water molecules to both parts of the expression obtained, we get Equation (3):4 NO + 4H_2_O ⇆ (2NO^+^ + 2OH^−^ +2H^+^) + (2NO^−^ + 2H^+^+ 2OH^−^)      2HNO_2_       2HNO(3)

Then, replacing 2HNO_2_ by (H_2_O + NO + NO_2_), and 2HNO by (H_2_O + N_2_O), we get Equation (4):4 NO + 4H_2_O ⇆ (H_2_O + NO + NO_2_) +2H^+^ + (H_2_O + N_2_O) + 2OH^−^(4)
from which Equation (2) characteristic of the gaseous phase follows.

The disproportionation reaction of NO molecules on the gaseous phase producing a significant amount of NO_2_ and N_2_O occurs at high pressure of NO—a few dozen atmospheres [74]. In water solutions the effectiveness of that reaction increases sharply if there are transition metal ions—Fe, Cu, Ni, Co, Mn, etc., that can catalyze the disproportionation of NO molecules—in the presence of the mentioned ions this reaction occurs at NO pressures above the solution of less than one atmosphere.

In this regard, the disproportionation reaction of NO molecules in water solutions catalyzed by Fe^2+^ ions characterized by high affinity to NO is especially effective [75]. Binding of two NO molecules with a Fe^2+^ ion in the presence of anion (L) ligands caused formation of relevant DNICs with molecular orbitals (MO), as combinations of 3d atom orbitals of iron and MO of NO and anionic ligands. Actually, including two NO molecules in the iron ligand sphere ensures that those molecules come closer to each other, which is sufficient for their disproportionation—transfer of an electron from one NO molecule to another one through iron d-electrons as “electron bridges” in accordance with the first reaction presented in Equation (5):   +(H_2_O, NO)[nL-Fe^2+^ (NO,NO)] ⇆ [nL-Fe^2+^ (NO^+^,NO^−^)]   →   [nL-Fe^2+^ (NO^+^,NO)] + HNO + OH^−^(5)

After hydrolysis of a nitroxyl anion that appears during disproportionation of NO molecules in the iron ligand sphere, the formed molecule of nitroxyl (HNO) leaves the iron ligand sphere with subsequent inclusion of the third NO molecule in it (Equation (5)). Via this procedure the iron dinitrosyl fragment achieves the resonance structure Fe^2+^(NO)(NO^+^) or, which is the same, Fe^+^(NO^+^)_2_, characterized by d^7^ electron configuration of iron or the formula{Fe(NO)_2_}^7^ as represented by Enemark-Feltham [68]. In this low-spin (S =1/2) form of iron dinitrosyl fragments, the corresponding DNICs, irrespective of the nature of anionic (L) ligands, yield the EPR signal within the range of g-factor values of 2.05–2.014, first registered at room temperature in the work by McDonald et al. [53]., and in our work at low temperature [54], the results of which are shown in Figure 4.

The hydrolysis of nitrosonium cations in iron dinitrosyl groups to nitrous acid (or nitrite anions), subsequently leaving the iron dinitrosyl group, initiates DNIC conversion into a corresponding mononitrosyl iron complex (MNIC) with anionic ligands, which has nL-Fe^2+^ (NO) resonance structure (Equation (6)):+H_2_O        [nL-Fe^2+^ (NO^+^,NO)] → [nL-Fe^2+^ (NO)] +HNO_2_/(H^+^+NO_2_^−^)(6)

Such complexes with water molecules included in them are quite well studied in the literature [76,77]. They are high-spin (S = 3/2) complexes with d^7^ electron configurations of iron, characterized by the EPR signal with g-factor values of g_x,y_~4.0 and g_z_ = 2.0 registered at a low temperature, shown in Figure 7d, on the right, registered in our work [76]. Besides, apart from that signal, there is a signal in the EPR spectrum characteristic of low-spin DNICs with water ligands, described by resonance structure [(nH_2_O)Fe^2+^ (NO^+^,NO)] with g = (2.05–2.012). Similar spectra which include EPR signals of high-spin MNICs [nL-Fe^2+^ (NO)] and low-spin DNICS with citrate, ascorbate or phosphate, are given in Figure 7b–d. In the solution of nitrosyl iron complexes with ethylene diamine tetraacetate (EDTA), only a mononitrosyl form of these complexes was found with the EPR method (Figure 7a).

Therefore, as a result of the disproportionation reaction of NO molecules in the ligand sphere of Fe^2+^ there may appear mono- and dinitrosyl iron complexes - [nL-Fe^2+^ (NO)] and [nL-Fe^2+^ (NO^+^,NO)] both with paramagnetic d^7^ electron configuration of iron, which, as represented by Enemark-Feltham, corresponds to the formulae: {Fe(NO)}^7^ and {Fe(NO)_2_}^7^. Comparing the concentration of these two types of nitrosyl iron complexes with phosphate in terms of integrated intensity of their EPR signals given in the work [78] showed that DNICs include no more than 20% of iron used for synthesis of these complexes and MNICs. The remaining iron was included in MNICs.

Another possible way could be an oxidative mechanism for DNIC formation, when these complexes include thiol-containing (RS-) compounds as anionic ligands; such compounds differ from the above-mentioned ligands, as they have high π-donor activity. Transfer of part of electron density from thiol sulfur atoms to iron atoms and nitrosyl ligands neutralizes the positive charge on these ligands, which sharply reduces the probability of their hydrolysis, i.e., binding of nitrosonium cations with hydroxyl ions. This provides for an exceptional stability of DNICs with thiol-containing ligands, i.e., complexes responsible for the 2.03 signal, discovered by us in the 1960s in living organisms [43,44,45,46]. Therefore, the mechanism for synthesis of these complexes can be represented as Equation (7)—which is a modified Equation (5):
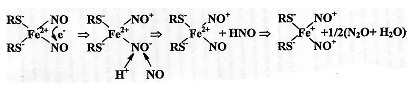
(7)

Blocking of nitrosonium cation hydrolysis in DNICS with thiol-containing ligands prevents their release from these complexes, due to which the latter cannot be converted into MNICs with the same thiol-containing ligands in accordance with Equation (6).

Formation of such MNICs was registered only at the initial stage of formation of DNICs with thiol-containing ligands [33,79,80], and MNICs ([(nRS^−^)-Fe^2+^-NO]) had a low-spin (S = 1/2) state and the EPR signal with the center at g = 2.04, with the triplet HFS registered at room temperature (Figure 8 and Figure 9).

Therefore, in my opinion, the oxidative mechanism for formation of DNICs with various anionic ligands considered above, caused by disproportionation of molecules in the ligand sphere of iron, provides an acceptable scheme for synthesis of both dinitrosyl and mononitrosyl forms of iron complexes with thiol-containing ligands in living organisms. As far as another, reduction-based mechanism for formation of these complexes is concerned, it has been recently proposed by Ford’s group [33,34,70,71] and seems to be less likely.

In accordance with this mechanism, at the initial stage of the reaction between Fe^2+^, NO and thiols, mononitrosyl iron complexes (MNICs) are formed, and they have Fe^2+^(NO)(RS)_2_ resonance structure, with a subsequent reduction of iron to a monovalent state by a thiol ligand and the release of a thiyl radical from the coordination sphere of iron. After this, the second NO molecule and a thiol-containing ligand in the Fe^+^ complex causes formation of a paramagnetic form of DNIC with two thiol-containing ligands with the EPR signal at g = 2.03 and {Fe(NO)_2_}^9^ configuration of the iron dinitrosyl group and, respectively, formally d^9^ electron configuration of iron [33,34,70,71].

The authors provided the following argument for the proposed DNIC formation mechanism: they register the appearance of thiyl radicals simultaneously with DNIC formation, using spin trapping [33,34,70,71]. However, as the authors did not manage to determine whether the numbers of produced thiyl radicals and DNICs were equal, the question regarding the mechanism for DNIC formation proposed in the works [33,34,70,71] remains unanswered. As regards thiyl radicals found by these authors, it will be shown in Section 7 that these radicals can appear as part of another DNIC formation mechanism not related to reduction of iron by thiols in MNICS.

The main counterargument to the mechanism for formation of DNICs with thiol-containing ligands proposed by P. Ford’s group is the following. This mechanism cannot account for the formation of paramagnetic mononuclear DNICs with non-thiol ligands, such as the above-mentioned DNICs with phosphate, hydroxyl anions or water, whose EPR signals are given in Figure 4 and Figure 7. Like DNICs with thiol-containing ligands, these complexes are low-spin-state compounds with S = 1/2 and EPR signals with g-factors close to g-factors of the 2.03 signal. However, if thiol-containing ligands which have strong reduction properties can provide reduction of iron dinitrosyl groups to{Fe(NO)_2_}^9^, such transformation can hardly be provided by phosphate or citrate anions as ligands in corresponding DNICs.

In the recent work by Ford’s group, its authors, while recognizing the counterargument to their mechanism for DNIC formation through reduction, assume that [Fe(NO)_2_]^9^ can appear as a result of disproportionation of [Fe^2+^(NO)] groups in accordance with Equation (8) [71]:2[Fe^2+^(NO)] →[Fe^+^(NO)_2_] + Fe^3+^(8)

This scheme does not provide arguments for transfer of the NO molecule from Fe^3+^ to Fe^+^, which ensures the appearance of the complex described by [Fe(NO)_2_]^9^. However, if we assume that disproportionation is related to iron dinitrosyl groups only, which appear in case of binding of NO molecules with divalent iron ions, this reaction can result in formation of complexes described as [Fe(NO)_2_]^7^ and [Fe(NO)_2_]^9^ (Equation (9)) or in the form of respective resonance structures, as shown in Equation (10):2[Fe(NO)_2_]^8^ → [Fe(NO)_2_]^7^ + [Fe(NO)_2_]^9^(9)
2[Fe^2+^(NO)_2_] → [Fe^2+^ (NO,NO^+^)] + [Fe^2+^ (NO,NO^−^)](10)

Both resonance structures in DNICs with non-thiol ligands cannot be stable due to hydrolysis of nitrosonium cations and nitroxyl anions included in them. Subsequent release of formed HNO_2_/(H^+^+NO_2_^−^) and HNO from these complexes should cause appearance of more stable MNICs ([nL-Fe^2+^ (NO)]) in the water solution in both cases.

A different situation is possible when these complexes have thiol-containing ligands. DNICs containing [Fe^2+^ (NO,NO^+^)] should remain, while DNICs containing [Fe^2+^ (NO,NO^−^)], like complexes with non-thiol ligands, should be converted into corresponding stable MNICs ([(nRS^−^)-Fe^2+^ (NO)]). This means that during disproportionation of DNICs with thiol-containing ligands with d^8^ electron configuration of iron (i.e., complexes described by Enemark-Feltham with the formula [Fe(NO)_2_]^8^) both dinitrosyl and mononitrosyl forms of these complexes should appear in the solution simultaneously, and in equal numbers. As we found out during our experiments that first MNICs and then DNICs with thiol-containing ligands were formed in the reaction of divalent iron, NO and thiols, we have no reasons to assume that these complexes can arise during disproportionation of DNICs with thiol-containing ligands with d^8^ electron configuration of iron, as shown in Equations (9) and (10).

As a final remark to this section, mononuclear EPR-active DNICs (M-DNICs) with thiol-containing ligands are capable of dimerizing in a reversible way in the living organisms as the content of thiol-containing compounds decreases, with formation of diamagnetic (EPR-silent) binuclear DNICs (B-DNICs) [81,82,83]. In accordance with their formula—[(RS)_2_Fe_2_(NO)_4_]—these complexes should be considered as Roussin’s red salt thioesters (named after the French chemist of the 19th century who synthesized this salt which is a dimer of two iron dinitrosyl groups bound with two inorganic sulfur atoms as bridges) [84].

We have reasons to assume that B-DNICs with thiol-containing ligands, where one of resonance structures can be represented as [(RS^−^)_2_Fe^+^_2_(NO^+^)_4_], like the similar structure in M-DNICs, and can also be donors of NO and NO^+^ in living organisms.

## 4. Can M- and B-DNICs with Thiol-Containing Ligands Really Be Donors of NO and NO^+^?

According to the previous section, in case DNICs are formed with thiol-containing ligands based on oxidation or reduction, which determine formally d^7^ or d^9^ electron configuration of iron in these complexes respectively, their resonance structures can be represented as [(RS^−^)_2_Fe^2+^ (NO,NO^+^)] or [(RS^−^)_2_Fe^2+^ (NO,NO^−^)]. Conforming to structures, chemical equilibrium between the complexes and their components is described in Equations (11) and (12) below:[(RS^−^)_2_Fe^2+^ (NO,NO^+^)] ⇆ Fe^2+^ +NO + NO^+^+ RS^−^ + RS^−^                NO^+^−RS^−^
(11)
(RS^−^)_2_Fe^2+^(NO,NO^−^)_2_ ⇆ Fe^2+^ +NO + NO^−^ + 2RS^−^(12)
showing the ability of DNICs with thiol-containing ligands obtained by oxidation or reduction to be donors of NO and NO^+^ or only NO and NO^−^, respectively.

Therefore, if DNICs with thiol-containing ligands can really be donors of NO^+^, this should be considered as solid evidence for the oxidative mechanism proposed by us and shown in Equation (7). Otherwise, we should consider that the mechanism causing formation of DNICs with d^9^ electron configuration of iron proposed by Ford’s group is true.

It is known that nitrosonium cations do not exist in water solutions alone: binding with hydroxyl ions, they are immediately hydrolyzed, converting into nitrite anions in case of neutral pH. Only when the solution contains thiol-containing compounds with higher affinity to nitrosonium cations than hydroxyl ions, they (in case of millimolar concentration) bind with these cations with formation of corresponding S-nitrosothiols (RS-NO) [40,85,86,87,88]. The latter are clearly identified by the optical absorption spectrum characteristic of them, with the band at 334 nm and extinction coefficient (ε) of 0.94 M^−1^ cm^−1^, and 40 times less intensive band at 543 nm [85].

Therefore, decomposition of DNICs with thiol-containing ligands with nitrosonium cation release should be accompanied by the formation of corresponding RS-NO which include thiol-containing ligands initially both not forming part of DNICs and forming part of them, as we said above. In case of oxidation of thiol groups of these compounds or their blocking by corresponding reagents during DNIC decomposition in case of neutral pH, nitrite anions should be accumulated as products of nitrosonium cation hydrolysis instead of RS-NO.

### 4.1. Transformation of Nitrosonium Cations into RS-NO at B-DNIC Decomposition

My experiments with stable DNICs with thiol-containing ligands, namely binuclear DNICs with glutathione (B-DNIC-GSH), proved what I stated above. I found out that decomposition of these complexes (if thiol groups of glutathione molecules released from B-DNIC-GSH remain intact) can be achieved by acidifying the solution of these complexes to pH 1.0–2.0 and subsequently incubating this solution to 80 °C for a short time. This caused quick protonation of thiol groups of glutathione which obviously provided for B-DNIC-GSH decomposition. In case of decomposition of these complexes via oxidation of thiol groups of glutathione, we found out that treating B-DNIC-GSH solutions with ferricyanide (FeCN) anions as one of the strongest oxidizers was sufficient enough [40,86,87,88].

B-DNIC-GSH decomposition was controlled based on changes in the optical absorption spectrum of these complexes—a decrease in intensity of absorption bands by 310 and 360 nm with ε of 4600 and 3700 M^−1^ cm^−1^ respectively (per one iron atom in binuclear complexes). As shown in Figure 10A, incubation of B-DNIC-GSH solution acidified to pH 1.0–2.0 0.4 mM at 80 °C for 30 s was sufficient for complete decomposition of these complexes, accompanied by formation of an equimolar (0.4 mM) amount of S-nitrosoglutathione (GS-NO) (Figure 10A, curve 4). Without incubation, B-DNIC-GSH remained unchanged in the acidified solution for at least 30 min (Figure 10A, curve 1–3).

The presence and absence of air (oxygen) did not influence B-DNIC-GSH decomposition leading to GS-NO (Figure 10B,C). However, when the temperature of the acidified B-DNIC-GSH solution was reduced from 80 to 40 °C, the time of decomposition of this complex and formation of GS-NO increased from 0.5 min to 26 min (Figure 10C,). Similarly, incubation of B-DNIC-GSH solution for a longer time (15 min) was necessary when B-DNIC-GSH concentration in the solution increased to 9 mM when the solution was heated at 80 °C in the presence of air (Figure 10D).

It is important that in all these experiments namely—in the presence and absence of air, at lower temperatures of B-DNIC-GSH solution or with increased concentration of this complex, during its decomposition, the corresponding S-nitrosothiol—GS-NO appeared in the concentration which was equimolar to that of B-DNICs (per one iron atom in these complexes). This means that in case of B-DNICs decomposition in the presence of acid, a half of nitrosyl ligands was released in accordance with Equation (11) in the form of nitrosonium cations participating in RS-NO formation.

It should be noted that in experiments shown in Figure 10, I used B-DNIC-GSH solutions where concentration of glutathione that did not form part of these complexes (“free glutathione”) was twice as high as that of the complexes. At the same time, I conducted experiments with incubation of B-DNIC-GSH solutions in one or two days after preparing these complexes which were kept in a fridge at 4 °C for that time in the presence of air. Special measurements showed that there was no significant decomposition of these complexes in such conditions. Nevertheless, keeping B-DNIC-GSH solutions in the presence of air could reduce the content of “free glutathione” in them through oxidating its thiol group with formation of a disulfide form of this tripeptide. Therefore, I aimed to check whether this reduction had an impact on the amount of GS-NO formed during incubation of acidified B-DNIC-GSH solutions in the presence and absence of air.

Experiments carried out to verify that showed that the above-mentioned treatment of B-DNIC-GSH solutions with double content of “free glutathione” in the presence of air in an hour after synthesis of these complexes did not cause significant decrease in concentration of GS-NO that appear during B-DNIC-GSH decomposition (no data is provided). However, similar experiments with keeping the same B-DNIC-GSH solutions in the absence of air showed a sharp decrease in GS-NO synthesis (Figure 11, curve 3) compared to the results shown on Panels B and C of Figure 10.

Certainly, there was a question, whether GS-NO formation increases during B-DNIC-GSH decomposition in the same experiments in the absence of air, but with lower initial concentration of “free glutathione” in the solution. I found out that in this case GS-NO formation increased sharply. When glutathione concentration decreased during B-DNIC-GSH synthesis to the level at which the concentration of the produced complex was equimolar to “free glutathione” concentration, the concentration of GS-NO formed during decomposition of B-DNIC-GSH in the absence of air became almost equal to the concentration of this complex (Figure 11, curves 4 and 5).

This result was obtained in similar experiments with solutions of B-DNIC-GSH with N-acetyl-L-cysteine (B-DNIC-NAC) (5 mM) with molar ratios of “free NAC” of 4: 1 or 1: 1 (Figure 12, curves 1 and 2, respectively). In those experiments, during decomposition of B-DNIC-NAC and increased concentration of “free NAC”, no S-nitroso-NAC (S-NO-NAC) was formed, even in a trace amount (Figure 12, curves 3 and 4). Meanwhile, S-NO-NAC formation during B-DNIC-NAC decomposition in the absence of air, with lower concentration of “free NAC”, the concentration of S-NO-NAC did not exceed 30% of the total amount of nitrosyl ligands in original B-DNIC-NAC (Figure 12, curves 5 and 6). A sharp decrease in RS-NO level observed in these experiments in the presence of an increased amount of free thiol in the solution was obviously related to the fact that thiols are capable of reduction of nitrosonium cations to NO instead of binding with them and forming corresponding RS-NO. As a result, both nitrosyl ligands were released from decomposing B-DNICs, obviously in the form of gaseous NO with neutral molecules.

This assumption was confirmed by experiments, where I managed to estimate the amount of gaseous NO released from B-DNIC-GSH during decomposition of these complexes in the presence of acid and in the absence of air. For that purpose I used the modified Thunberg apparatus with a welded cylindrical cuvette with quartz sides (Figure 13A), in which optical absorption of gaseous NO was registered. To accomplish this, the cuvette was installed in the spectrometer along the ray.

The experiments were conducted in the following way. In the upper part of the apparatus with the total volume of 120 cm^3^, 17 mL of 12 mM B-DNIC-GSH solution (i.e., 200 μmol of complexes per one iron atom) was placed, with free glutathione content that was three times higher than the content of B-DNICs, and in the lower part we introduced 0.1 mL of 100% sulfuric acid. After pumping air out of the apparatus and its heating to 80 °C, acid was added to B-DNIC solution, which immediately caused bubbles of gaseous NO released from B-DNICs, as shown in Figure 14, for about one hour. By that time, ~400 μmol of NO was found in the gaseous phase, i.e., all nitrosyl ligands (two per one iron atom in B-DNICs) were released from these complexes in the form of NO.

In similar experiments with B-DNIC-GSH solutions that contained free glutathione in the amount equimolar to the amount of iron nitrosyl fragments in these complexes, the level of nitrosyl ligands released in the form of NO molecules considerably decreased (no data is provided).

Therefore, it is only natural to state that absence of corresponding RS-NO among products of B-DNIC decomposition with glutathione or NAC ad with increased content of the free form of these thiols in the solution was related to the fact that they provided for reduction of nitrosonium cations which could be released from these complexes. The latter occurred only when thiol amount in the solution was small.

### 4.2. Transformation of Nitrosonium Cations Released from DNIC into Nitrite Anions

Presence of nitrosonium cations in B-DNIC-GSH could be detected by the accumulation of nitrite anions in the solutions of these complexes during B-DNIC decomposition resulting from one of the strongest oxidizers, potassium ferricyanide (FeCN).

This was confirmed when the solution was acidified to pH 1–2 with a subsequent release of gaseous NO (Figure 15, curves 4 and 5), obviously due to disproportionation of the formed HNO_2_ molecules. In accordance with this reaction, about 110 mM of the total amount of nitrosyl ligands in B-DNIC-GSH equal to 260 mM were converted into NO. This means that about 70% of the assumed number of nitrosonium cations in the complexes were converted into NO. Therefore, decomposition of B-DNIC-GSH related primarily to oxidation of thiol groups due to FeCN caused release of NO and NO^+^ from them in almost equal amounts.

I got a similar result when I incubated the aqueous solution of B-DNIC-GSH at pH 7.3 in the presence of air at 80 °C for about one hour. It resulted in a decomposition of the complex due to oxidation of thiol groups in thiol-containing ligands (Figure 16, curve 2). As I had assumed, this also led to nitrite formation due to hydrolysis of nitrosonium cations released from B-DNICs. Subsequent acidifying of this solution to pH 1–2, caused by adding glutathione to it, led to converting nitrite into HNO_2_, and therefore, conversion of glutathione into GS-NO. Concentration of the latter was equal to the concentration of B-DNIC-GSH—0.5 mM (per one iron atom in the complexes), implicating that a half of nitrosonium cations was released from the complexes in the form of NO^+^.

In similar experiments conducted by Hogg’s group, keeping 0.1 mM of B-DNIC-GSH solution in the presence of air at neutral pH led to the accumulation of nitrite (only nitrite) in the same concentration of 0.1 mM, which emerges via hydrolysis of nitrosonium cations released from B-DNICs after decomposition of complexes, obviously due to the oxidation of the thiol group in glutathione [89]. Of note is GS-NO, the product of reaction between nitrosonium and glutathione that appeared in case of glutathione presence, but then disappeared, being involved in the reverse equilibrium reaction between B-DNICs and its constituent components (Figure 1) [40]. As the thiol group in glutathione was oxidized, more and more nitrosonium cations got involved in hydrolysis with nitrite production, which made B-DNIC rearranging reaction irreversible. This process finished with the conversion of all nitrosonium cations released from B-DNICs into nitrite anions.

I demonstrated the conversion of nitrosonium cations into nitrite in the experiments with nitrosyl iron complexes with water [87,88]. As I mentioned in Section 3, these complexes exist in two forms, high-spin MNICs-H_2_O with S = 3/2 and low-spin MNICs—H_2_O with S = 1/2, whose EPR signals with the main absorption at g = 4.0 and g_._ = 2.03, respectively, are shown in Figure 7e. These complexes are unstable, provided that they quickly decomposed (within 20–30 min) when NO was pumped out of the Thunberg apparatus, where these complexes were synthesized. This decomposition (at neutral pH of the solution), accompanied by a release of divalent iron ions into the solution, caused turbidity in the solution due to inclusion of these ions into insoluble hydroxide complexes. When the solution was then acidified to pH 1–2 as a result of adding glutathione to it in the amount which was 50 times exceeding the amount of iron, the solution was becoming clearer, which allowed resisting the absorption spectrum in it with the main and weak absorption bands at 334 and 543 nm respectively, which is characteristic of S-nitrosoglutathione (GS-NO) (Figure 17, curves 2, 3 [87,88].

I have strong reasons to assume that the appearance of GS-NO was related to the reaction of glutathione with nitrite, converted into nitrous acid in the acid medium. Nitrite most naturally appeared via hydrolysis of nitrosonium cations released from decomposing nitrosyl iron complexes, namely from M-DNIC-H_2_O. As far as high-spin M-DNIC-H_2_O are concerned, their decomposition should not have caused the appearance of nitrosonium cations in the solution, which would be then converted into nitrite. This is supported by the fact that that nitrosyl iron complexes with EDTA which were presented only by high-spin MNIC-EDTA (Figure 7a), did not produce GS-NO during decomposition in the presence of glutathione.

It should be noted that gaseous NO used in these experiments was characterized by the presence of NO_2_ as impurity, and therefore N_2_O_3_ as impurity, capable of S-nitrosation of thiols. This was proved by the appearance of optical absorption of GS-NO in the solution after its treatment with gaseous NO in the absence of exogenous Fe^2+^ with a subsequent addition of glutathione to the solution (Figure 17. curves 1 and 1a). At the same time, the concentration of produced GS-NO was much lower than that in the solution in the presence of added iron (Figure 17, curves 2 and 3).

### 4.3. Decomposition of B-DNIC-GSH Caused by N-methyl-D-Glucamine Dithiocarbamate (MGD)

Compounds that can ensure an effective decomposition of M- and B-DNICs with thiol-containing ligands include derivatives of dithiocarbamate (R_2_ =NCS_2_^−^)-N-methyl-D-glucamine dithiocarbamate (MGD) or diethyldithiocarbamate (DETC) [90,91,92]. Having high affinity to the iron-mononitrosyl group, these compounds can, as shown in Figure 2, intercept these groups from iron-dinitrosyl fragments in M- and B-DNICs with formation of EPR-registered MNICs with dithiocarbamate ligands and release of nitrosonium cations, binding with various thiol-containing compounds, from M- and B-DNICs:

Such a transformation of B-DNIC-GSH in case of its contact with MGD, which is obviously caused formation of S-nitrosated MGD derivative, was shown in our work [38]. Optical identification of a volatile mixture released from B-DNIC-GSH + MGD solution by means of Thunberg apparatus shown in Figure 13A showed absence of any NO traces, and only release of carbon disulfide from the excessive amount of MGD (100 mM), gradually decomposing due to hydrolysis (Figure 18, the optical absorption band at 204 nm) [94]: Concerning the nitrosyl ligands forming part of B-DNIC-GSH, half of them was found in the form of NO in MNIC-MGD, which gave the triplet EPR signal shown in Figure 18, Inset.

As far as the second half of nitrosyl ligands in B-DNICs is concerned, it was obviously bound in the form of NO^+^. Subsequent acidifying to pH 1–2, which induced decomposition of the whole MGD as a result of its hydrolysis with release of a considerable amount of carbon disulfide in the gaseous phase (Figure 18, curve 2) also caused the appearance of gaseous NO released from decomposed MNIC-MGD and S-nitroso-derivative of MGD. Its amount was equal to the number of nitrosyl ligands in B-DNIC-GSH placed in the Thunberg apparatus.

Therefore, these experiments also showed the presence of nitrosonium cations in these complexes, released from them during their decomposition with the impact of MGD. This is fully consistent with our ideas of d^7^ electron configuration of iron in these complexes and their [(RS^−^)_2_Fe^+^(NO^+^)_2_] resonance structure.

## 5. {Fe(NO)_2_}^7^ Representation of the Iron-Dinitrosyl Unit (or Formally d^7^ Electron Configuration of Iron in That Unit), in DNICs with Thiol-Containing Ligands as a Real Characteristics of These Complexes in the Solution

In the previous sections I did not emphasize that all information in those sections was related to the behavior and properties of DNICs with thiol-containing ligands in their aqueous solutions. Meanwhile, if we consider contemporary concepts concerning the electronic and spatial structure of these complexes based on the results of X-ray analysis of DNICs with thiol-containing ligands in the crystalline state, we will see that there is no match between these results and results of our research of these complexes in the solution. The X-ray analysis showed that in the crystalline state, DNICs have tetrahedron spatial structure [64,65,66], so the low-spin state with S = 1/2 for M-DNICs characterized by the 2.03 signal can be achieved in case of iron localization of nine electrons on 3d orbitals, or more precisely, on the corresponding MO of complexes that include atom orbitals of divalent iron ions [78]. Therefore, d^9^ electron configuration of iron in DNICs is more likely (and most researchers consider it to be true) than our idea of d^7^ variant.

This controversy can be resolved only if we assume that X-ray characteristics of DNICs are not applicable to these complexes in the solution. In other words, when DNIC crystals are dissolved, primarily in water, the structure and, consequently, chemical properties of DNICs with thiol-containing ligands change in case of interacting with water molecules. Due to this, these complexes can become donors of nitrosonium cations, which we have already shown.

Our idea about d^7^ electron configuration of iron in DNICs with thiol-containing ligands and the corresponding ([RS^−^)_2_Fe^2+^(NO,NO^+^)] resonance structure can also be proved by the results of analyzing the characteristics of the 2.03 signal of these complexes.

The values of the g-factor and the HFS components of ^57^Fe in the EPR signals of M-DNIC with L-cysteine, glutathione, ethyl xanthogenate, and thiosulfate determined in our studies [67,95,96,97] are as follows: g_⊥_ = 2.040, g_‖_ = 2.014 (g_aver_. = 2.03), A_⊥_(^57^Fe) = −1.7 mT, A_‖_(^57^Fe) = −0.25 mT, A_iso_(^57^Fe) = −1.22 mT (DNIC with L-cysteine or glutathione) [39,78,95]; g_⊥_ = 2.042, g_‖_ = 2.014 (g_aver_. = 2.033), A_⊥_(^57^Fe) = −1.7 mT, A_‖_ (^57^Fe) = −0.35 mT, A_iso_ (^57^Fe) = −1.25 mT (DNIC with ethyl xanthogenate [97] g_⊥_ = 2.045, g_‖_ = 2.014 (g_aver_. = 2.035), g_⊥_ = 2.045, g_‖_ = 2.014, A_⊥_ (^57^Fe) = −1.7 mT, A_‖_ (^57^Fe) = –0.3 mT, A_iso_(^57^Fe) = −1.25 mT (DNIC with thiosulfate) [95]. As can be seen, in all these cases g_⊥_ > g_‖_ > 2.0023 and |A_⊥_ (^57^Fe)| > |A_‖_ (^57^Fe)|.

The same ratio between g_⊥_ and g_‖_, and between |A_⊥_ (^57^Co^2+^)| > |A_‖_ (^57^Co^2+^)| was established for the EPR signal of low-spin ^57^Co^2+^ complexes with phthalocyanine and porphyrin ligands with d^7^ electron configuration, described and theoretically calculated in McGarvey’s work [98]. Moreover, this work established that with the mentioned ratio between g-factor values and HFS tensor components for ^57^Co (and consequently ^57^Fe), the unpaired electron should be on d_z_2 orbital of the metal. This conclusion is applicable to DNICs with thiol-containing ligands in case of planar-square spatial structure of these complexes with d^7^ electron configuration of iron and a diagram of antibonding molecular orbitals shown in Figure 19, which was described in [95,96]:

As another experimental fact proving that the concept of d^7^ electron configuration of iron in DNICs with thiol-containing ligands is true, we can mention the results of our studies related to reduction of these complexes [99,100]. We found out that reducing both M-DNICs and B-DNICs with glutathione based on the two-electron mechanism to the paramagnetic state with S = 1/2 and d^9^ electron configuration of iron is possible, which is entirely consistent with the idea. This result is shown in Figure 20, where a change in the absorption spectra of these complexes in case of their treatment with sodium dithionite, a strong reducing agent, is given:

B-DNIC-GSH was synthesized with the ratio Fe^2+^: GSH = 1: 10 at pH 7.4, while M-DNIC-GSH were obtained by increasing pH of B-DNIC-GSH to 10.7. At the same time, instead of absorption bands of B-DNIC-GSH at 310 and 360 nm (Figure 20, curve 1), a band at 390 nm appeared (Figure 20, curve 2), which is characteristic of M-DNICs. After sodium dithionite treatment, a band at 650 nm appeared in the spectra of B- and M-DNIC-GSH, which is characteristic of a reduced paramagnetic form of these complexes. This change of spectra was fully reversible to curves 1 and 2 in case of oxidation of the paramagnetic form of the complexes with air.

For both types of DNICs, the same EPR signals with g_⊥_ = 2.01 and g_‖_ = 1.97 at 77 K, and singlet signals at g = 2.0 at room temperature were registered (Figure 21):

The EPR-active form of reduced M-DNIC-GSH did not appear when glutathione content in the solution was 200 times higher than that of M-DNICs. When sodium dithionite was added, only the 2.03 signal characteristic of M-DNICs disappeared.

Given that d^7^ electron configuration of iron in M-DNIC-GSH is true, the latter could be accounted for the glutathione excess in the solution, when this complex can be reduced only to the EPR-silent d^8^ electron configuration of iron, while a lower concentration of GSH provides EPR-active and d^9^ configurations. Due to a low content of free glutathione (not being part of the complexes) the reduction of iron-dinitrosyl fragments in B-DNIC-GSH, the fragments are first converted into d^8^, and then into the paramagnetic d^9^ configuration (mechanism of this conversion is described in detail in [101]).

If we consider the initial d^9^ electron configuration of iron M-DNICs, which is due to their formation by reduction, these complexes should not be converted into the EPR-detectable form in case of reduction. Accepting one electron, they should already be converted into the diamagnetic d^10^ electron configuration corresponding to {Fe(NO)_2_}^10^ representation of the iron-dinitrosyl group.

Thus, the data considered here allow to propose that {Fe(NO)_2_}^7^ representation of the iron-dinitrosyl unit corresponding to formally d^7^ electron configuration of iron in that unit in DNICs with thiol-containing ligands is a real characteristics of these complexes in the solution.

## 6. Synthesis of M-DNIC with Thiol-Containing Ligands Based on S-Nitrosothiols

In the previous sections of my review I did not mention a mechanism behind DNICs formation with thiol-containing ligands in the reaction of Fe^2+^, thiols and S-nitrosothiols, which we discovered in the 1990s and studied it in early 2000s [69,80,101,102]. It is shown in Equation (13):
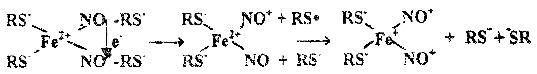
(13)

In accordance with this mechanism, at its first stage, binding of Fe^2+^ in thiol surrounding with two S-nitrosothiol molecules represented by their resonance structures NO-RS^−^ and NO^+^-RS^−^ occurs. As a result of a subsequent disproportionation reaction between these structures, unstable and quickly decomposed compounds NO^+^-RS^−^ and NO-RS^−^ appear, which leads to the formation of M-DNICs with [(RS^−^)_2_Fe^+^(NO^+^)_2_] resonance structure.

In accordance with Equation (13), Borodulin, an employee of our laboratory, developed a simple way of synthesizing DNICs with glutathione or cysteine, which is available to any researcher related to chemistry [101]. This consists in adding thiols first, then a divalent iron salt, and only after that sodium nitrite to distilled water. The following processes take place in this reaction. Thiols (glutathione or cysteine) reduce pH of the solution to “acid” values, due to which the iron salt is dissolved in an ion form without producing hydroxide sediment. When nitrite is then added to the acid solution, nitrite undergoes protonation and forms RS-NO with thiol, indicated visually by the appearance of a pink color and in UV/Vis spectra by an absorption band at 334 nm. Then pH of the solution increases to neutral values, inducing the formation DNICs with thiols (mainly its B-form) in some time in accordance with Equation (13). The level of the latter is controlled optically based on the intensity of the absorption band of this form at 360 nm.

As a particular example of such synthesis, I would like to provide a protocol for obtaining 2.5 mM B-DNIC with glutathione in an aqueous solution. The following is added to 10 mL of distilled water: 62 mg of glutathione (20 mM), then 28 mg of FeSO_4_ × 7H_2_O (10 mM), and after that 6.9 mg of sodium nitrite (10 mM). After adding the first tow reagents, pH of the solution decreased to 3.8. After adding nitrite, the solution became pink, which served the evidence for GS-NO formation. Optical absorption intensity of GS-NO (in the band at 334 nm) shows that formation of this compound being limited by the amount of nitrite (10 mM) finished in 1.5–2 h. After that, pH of the solution was increased to 7.2 by adding drops of saturated NaOH solution, proceeding in accordance with Equation (13) to the formation of M-DNICs with a subsequent conversion into B-DNICs. Completion of this process took several hours. For instance, it is sufficient to keep the solution in the refrigerator for a night. During this process, only a half of the iron (5 mM) was included in DNICs, and the remaining iron formed residue of hydroxide complexes during the night. To remove this residue, the solution was filtered using filter paper. Subsequent analysis of the obtained solution using methods such as optical spectrophotometry and high-performance liquid chromatography (HPLC) on a silica gel column modified by an anion detergent (cetyltrimethylammonium chloride) showed that the obtained solution contained 5mM B-DNICs (per one iron atom in the complex) and 1 mM of oxidized glutathione. Nitrite and GS-NO were found in the solution in trace amounts.

In the given protocol for synthesizing DNICs, 10 mM of nitrite was used. It provided for formation of the same amount of GS-NO that was twice as large as iron concentration in DNICs. This is entirely consistent with the mechanism for M-DNIC formation shown in Equation (13): in accordance with it, there are two GS-NO molecules per each iron ion.

It should be noted that in accordance with Equation (13), synthesis of M-DNICs with participation of S-nitrosothiols should be accompanied by appearance of thiyl (RS^−^) radicals in the solution. It should not be ruled out that these radicals could be found in the above-mentioned work by Ford’s group [33,34,70,71]. As we indicate in the next section, processes of DNIC formation with participation of gaseous NO and RS-NO are closely related, and as a result, the first process is changed to the second one, which can cause appearance of thiol radicals in accordance with Equation (13).

## 7. Oscillatory Interconversions of DNIC and S-Nitrosothiols

The ability of S-nitrosothiols to provide (in accordance with Equation (13) for DNIC formation, as well as the ability of the latter as donors of nitrosonium cations to provide (in accordance with Equation (7) for conversion of thiols into S-nitrosothiols serves as solid evidence that NO, thiols, S-nitrosothiols (RS-NO) and DNICs are closely related, which is shown in Figure 3. This scheme demonstrates the equilibrium processes (horizontally), which characterize the ratio between B- and M-DNICs, and between M-DNICs and their components. At the same time, the vertical shows reactions of M-DNIC synthesis involving NO or RS-NO. Both reactions are irreversible, as they are accompanied by the formation of nitrous oxide or disulfide, which cannot get involved in reverse reactions:

In accordance with the scheme, in case of excess of NO or RS-NO in the solution containing divalent iron and thiols, M-DNICs can be synthesized in accordance with reaction 1 or reaction 2 that are autocatalytic. During these reactions interconversion of NO into RS-NO or RS-NO into NO respectively should occur. As a result, reaction 1 should be changed to reaction 2 due to RS-NO accumulation, and vice versa, reaction 2 should be changed to reaction 1 due to NO accumulation. Given a sufficiently high level of NO and thiols, which are converted into nitrous oxide and disulfide respectively, it is maintained in this system, and therefore, their amount is reduced, the system of chemical processes shown on Figure 3 can support itself for a relatively long time. Actually, this system can be maintained through the energy of the reaction of oxidation of thiols by NO, which causes accumulation of disulfides (RS-SR) and nitrous oxide (N_2_O) respectively.

The autocatalytic “two-step” mechanism for M-DNIC formation based on non-equilibrium reactions 1 and 2 shown on Figure 3 with different constants of such reaction speed [80] allows assuming (in accordance with I. Prigogine’s idea about the role of autocatalysis in establishing oscillatory and autowave modes of chemical reactions [103]) that such mode is possible for the system of chemical reactions shown on Figure 3. This is consistent with the results of our experiments to launch synthesis of M-DNIC-GSH by applying a spherical drop (0.01 mL) of the aqueous solution of Fe^2+^ (10 μM) + GSH (0.5 M) on a thin (0.3 mm) layer of 0.5 mM aqueous solution of GS-NO. Then we photographed processes taking place in the GS-NO layer once in 40 ms [104].

According to Figure 22, left panel, contact of the drop containing Fe^2+^ + GSH with GS-NO solution caused a complex picture of ring-like structures with a maximum diameter of ~10 mm in the solution in 40 ms already, which was caused by M-DNIC formation process. Obviously, later, the diffusion of ring components caused distortion of these structures, so in 200 ms the number of rings decreased considerably, and in five seconds the ring-like structure completely disappeared.

The ring-like structure also appeared after adding a drop of nitrite solution to GS-NO solution. As there was no chemical process in this case, such structure was obviously related to mechanically induced waves. As distinct from the complex and quickly disappearing ring-like structure shown on the left panel of Figure 22, the ring-like structure related to mechanically induced waves remained unchanged for at least 1–2 s.

Therefore, it should not be ruled out that in case of mixing a drop containing FeSO_4_ + glutathione with GS-NO solution that caused formation of M-DNIC-GSH, two types of wave processes could occur: the first process, the mechanically induced one, related to submerging the 10 μL spherical drop of the mixture of 1mM FeSO_4_ + 0.5M GSH in GS-NO solution, and the second process, related to the autowave nature of the process for formation of the above-mentioned complexes. Overlapping of autowaves and mechanically induced waves could cause a sharp increase in the number of ring-like structures, which were quickly distorted and disappeared, obviously due to Brownian (random) diffusion of colored products of the reaction between M-DNICs and GS-NO.

The autowave picture in our experiments was initiated by contact of the solution of Fe^2+^ + glutathione with the solution of GS-NO, leading to M-DNIC formed with glutathione in the non-equilibrium reaction 2 shown on Figure 3. Subsequent reduction in GS-NO level in the reaction mixture accompanied by accumulation of NO molecules in it could cause formation of M-DNICs in accordance with non-equilibrium reaction 1, etc., with subsequent fading of this process and establishing chemical equilibrium between M-DNICs and their components. It is obvious that a change of non-equilibrium reactions 1 and 2 could determine autowave processes in the chemical system shown on Figure 3.

If we find out that the chemical system shown on Figure 3 is capable of the autowave mode in dynamics, this can have fundamental implications. It will be possible to state that functioning of NO and its endogenous compounds, DNICs with thiol-containing ligands and RS-NO, arises from autowaves rather than from free diffusion of these compounds in the low-molecular-weight form in the intercellular space. In other words, instead of random Brownian motion of these agents, there is some directional and more effective transfer of them to their biological targets.

## 8. Biological Activity of DNICs with Thiol-Containing Ligands as Donors of NO and NO^+^

Physical and chemical studies of DNICs with thiol-containing ligands considered in the previous sections of this review lead mainly to the following biological insights. The mechanism for DNIC formation proposed in course of these studies is based on their synthesis from the “building materials” such as neutral NO molecules, Fe^2+^ ions and thiols. The mechanism is determined by disproportionation reaction of NO molecules related to the acid-base reaction—protonation of nitroxyl anions (NO^−^) which arise during disproportionation of NO. The produced M-DNICs include equal numbers of neutral NO molecules and their ionized form—nitrosonium cations (NO^+^). Therefore, one of resonance structures of these complexes can be presented as [(RS^−^)_2_Fe^2+^ (NO,NO^+^)]. Therefore, M-DNICs with thiol-containing ligands, like their binuclear analogues, can be donors of both NO and NO^+^.

We had claimed this earlier based on experiments conducted with solutions of DNICs with thiol-containing ligands [80,86,87,88]. However, it has not been widely recognized yet. The presence of nitrosonium cations in M- and B-DNICs with thiol-containing ligands, like in the same complexes with non-thiol ligands, is considered to be highly unlikely by many researchers being our opponents in this question. S-nitrosation reactions or reactions leading to accumulation of nitrite anions, detected in solutions of these complexes, are usually presented as an artefact. It is assumed that due to the uncontrolled access of air to DNICs solution, NO which might be released from these complexes could be oxidized in these conditions to nitrogen dioxide with a subsequent formation of nitrogen trioxide—the donor of NO^+^, which could bring about S-nitrosothiols or nitrite in DNIC solutions [89].

Meanwhile, several years prior to these statements, Lancaster et al. conducted experiments with cell cultures and showed that the appearance of S-nitrosothiols in those cultures correlated with DNICs formation and did not depend on the level of oxygen in cells [36]. Besides, earlier, we had demonstrated in our joint work with German researchers that there was S- and N-nitrosation of proteins initiated by DNICs with cysteine which did not depend on oxygen [105]. Obviously, those results were less convincing for our opponents. Therefore, in my opinion, it is necessary to consider the results of our recent studies of human tumor cells in greater detail as they provide evidence in favour of our hypothesis. We have demonstrated that DNICs with thiol-containing ligands have a cytotoxic effect on these cells due to release of components from these complexes, such as nitrosonium cations.

These studies were inspired by the results of Russian and German research, conducted in Germany, where it was shown that Bcl-2-independent pro-apoptotic effect of DNICs with thiosulfate on human Jurkat leukemia cell line occurred [106]. At the same time, it was found out that in case of adding MGD (in one minute after M-DNICs) to these cells, which caused MNIC-MGD appearance instead of M-DNICs in the same concentration, the number of pro-apoptotic cells increased, exceeding the total effect of DNICs and MGD taken separately (Figure 23):

The increase in the pro-apoptotic effect of M-DNIC-thiosulphate in the presence of MGD (Figure 23, column 4) is explained by the authors [106] in compliance with the mechanism for interaction between B-DNIC-GSH and MGD, which I described in Section 4.3. Due to this interaction, a half of nitrosyl ligands forming iron-mononitrosyl group, is transferred from DNICs to MNIC-MGD characterized by the corresponding triplet EPR signal, while another half is released in the form of free nitrosonium cations (Figure 2). According to the authors [106], this half caused S-nitrosation of cell surface proteins of Jurkat cells, which initiated apoptosis of these cells.

Regarding the weakening of apoptotic effect of M-DNIC-thiosulfate in the presence of a considerable glutathione amount (Figure 23, column 3), it could be caused by conversion of M-DNIC-thiosulfate into stabler and poorly decomposing DNIC-GSH, i.e., complexes which weakly generate free nitrosonium cations in the solution.

We have recently got similar results on MCF7 human breast cancer cells and bacterial Escherichia coli TN530 cells during their treatment with B-DNICs with mercaptosuccinate (MS) or B-DNIC-GSH and then MGD or another dithiocarbamate derivative—diethyldithiocarbamate (DETC) respectively. The results of these studies are given in Figure 24 [93].

On the left panel of Figure 24 the differences between 2D plots of AnnexinV fluorescence (marker of apoptosis) vs. Propidium iodide fluorescence (marker of dead cells) of MCF7 cell suspensions incubated with B-DNIC-MS with or without MGD for 48 h are shown. These data are summarized in the following diagram (Figure 24, right panel). It demonstrates that the cytotoxicity of B-DNIC-MS at the combined treatment of MCF7 cells with B-DNIC-MS and MGD sharply increases. Similar diagrams showing a decrease in MCF7 cell viability according to MTT assay were obtained in experiments with B-DNIC-MS and B-DNIC-GSH when MGD was added to the latter [93].

Therefore, results of our experiments [93], like those of the experiments described in [106], are fully consistent with our ideas that DNICs with thiol-containing ligands can be sources of nitrosonium cations as agents causing both animal and bacterial cell death. It should be noted that a similar conclusion regarding Roussin’s black salt mentioned at the end of Section 3 (NH_4_[Fe_4_S_3_(NO)_7_]), and Roussin’s red salt ester [Fe_2_(SCH_2_CH_2_OH)_2_(NO)_4_] was made by the group of Cammack, based on their experiments with *Clostridium sporogenes* in 1992 [107].

Another possibility should not be ruled out as well. Nitrosonium cations as DNIC components can be responsible for the ability to suppress the development of experimental endometriosis in rats, as we discovered [19,20,108,109,110,111]. These studies, which were very difficult to conduct, started in our laboratory in 2011–2012 when my collaborator Evgenia Burgova managed to reproduce the method for initiating experimental endometriosis in rats. It consisted in surgical transplantation of two autologous fragments (2 × 2 mm) of uterine tissue (the endometrium together with the myometrium) excised from the left uterine horn onto the anterior surface of the abdominal wall. Thirty to thirty-five days after surgery, the implants developed into large-size (~1 cm) oval-shaped endometrioid tumors (EMT) (Figure 25A,B); their growth stopped gradually within two months after surgery.

Four days after the surgery, the animals were given an intraperitoneally dose of B-DNIC-GSH (~10 μmol/kg as calculated per one iron atom in B-DNIC). The treatment course included one daily injection of B-DNIC and lasted 10 days with a subsequent two-week standard vivarium diet. After completing the treatment course, EMT failed to be detected in the majority of experimental rats (Figure 25C), while in the control group large-size EMT (mean volume ≤ 100 mm^3^) continued to develop. Secondary EMT were found in some animals from the control group, as shown in Figure 25B, arrows 3–5.

To prove the role of nitrosonium cations in the cytotoxic effect of B-DNIC-GSH on EMT, experiments to study the impact of dithiocarbamate derivatives on EMT development with B-DNIC impact on them are considered to be promising. These studies can open up the way to create promising drugs for treating endometriosis, which is frequently found in women now, and is a serious and socially dangerous disease.

In 2002, a work by German researchers was published [112], which showed that M-DNICs with cysteine themselves, and especially connected with the cysteine residue of tetrapeptide Ley-Ser-Tre-Cys, partially binding with the 2A protease of Coxsackie-B virus, completely suppressed the activity of this enzyme. Such inhibition observed in experiments with both isolated protease and protease included in cell cultures was related to S-nitrosation of one of thiol groups of this enzyme, and it was completely reversible. When the S-nitrosated 2A protease was treated with dithiotreitol that intercepted nitrosonium cations, the activity of the protease was completely restored. Similarly, M-DNICs with cysteine themselves initiated S-nitrosation of intracellular caspase-3 [112]. It is pointed out that S-nitrosation of the 2A protease in mouse myocardium tissues considerably weakened infection of mice with Coxsackie-B virus [113].

According to the authors [114], the ability of M-DNICs with cysteine to cause S-nitrosation of the 2A protease was related to the fact that both nitrosyl ligands in iron dinitrosyl groups of those complexes existed in the form nitrosonium cations, which caused S-nitrosation of the protease. This is only partially true. Our data show that DNICs with thiol-containing ligands are donors of neutral NO molecules, usually released from DNICs in the same amount as nitrosonium cations.

Based on these data, we have valid reasons to assume that DNICs with thiol-containing ligands, as compounds capable of S-nitrosation of proteases in hosts and viruses in case of viral infections, can inhibit SARS-Cov-2 infection both in experiments in vitro (with cell cultures) and in vivo (with animals) [114]. Once this assumption is confirmed, its verification can form the basis for developing a drug for COVID-19 treatment. It is obvious that using DNICs with thiol-containing ligands for this purpose can be most effective in case of inhaling sprays (produced by a nebulizer) of aqueous solutions of these complexes. DNICs which easily penetrate epithelial cells in the respiratory tract and lungs can prove to be most efficient. Therefore, DNICs with c N-acetyl-L-cysteine can be recommended; according to the recent data, they are even capable of penetrating the skin [115].

Recently, Darensbourg and her colleagues from Texas A&M University published a paper where they demonstrated that B-DNICs with thiol-containing ligands can act as inhibitors of the SARS-CoV-2 main protease due to iron-dinitrosyl unit binding with thiol groups of the protease [116].

If the ability of DNICs with thiol-containing ligands to be the source of nitrosonium cations determines the cytotoxic effect of these complexes, the fact that they release neutral NO molecules provides for their positive and regulatory function in animal and human organisms. This is related to strong vasodilating action of DNICs with various thiol-containing ligands, and consequently, dose-dependent hypotensive effect of these complexes [1,2,3,4,5,6,7,8,9]. Currently, we have collaborated with the National Medical Research Center of Cardiology of the Ministry of Health of the Russian Federation to create and start commercial production of Oxacom, a hypotensive drug that includes B-DNIC-GSH as an active ingredient. The drug is undergoing clinical trials. The strong hypotensive effect of this drug lasting for several hours after one-time administration in volunteers is proved by the data shown in Figure 26 [7]:

As DNICs with thiol-containing ligands functioning are NO donors, they inhibit platelet aggregation, which is more effective (dose-dependent) than sodium nitroprusside, which is famous due to its thrombolytic action [10,11,12,13,15]. As NO donors, DNICs have a similar positive effect on healing various types of wounds, including diabetic ulcers [17]. Interesting results have been obtained in animals: DNICs action impacts the penile erectile function [16]. We should not rule out the possibility of creating drugs based on these complexes which can be used for treatment of erectile dysfunction in humans as well.

Our group has been studying the biological activity of DNICs with thiol-containing ligands for over 20 years, with subsequent practical use of the results in medicine. We have reported these results in numerous articles and two monographs, presented in References, but I still would like to mention the recently published work by scientists from Taiwan, where they provide data on the ability of low-molecular-weight DNICs with thiol-containing ligands to ensure NO penetration in brain tissues through mediation of proteins such as serum albumin and gastrointestinal mucin, activating hippocampal neurogenesis and ameliorating the impaired cognitive ability. Administration of low-molecular-weight DNICs was oral [28].

## 9. Conclusions

The mechanism for formation of DNICs with various anionic (L) ligands (including thiol-containing ligands) we have proposed is based on the disproportionation reaction of two NO molecules binding with a divalent iron ion. It ensures the formation of low-spin DNICs characterized by the presence of one neutral NO molecule and one nitrosonium cation (NO^+^) in the iron-dinitrosyl fragment of these complexes. The corresponding resonance structure of a mononuclear form of DNICs is described as [(L^−^)_2_Fe^2+^(NO)(NO^+^)]. In the absence of thiol-containing ligands this structure is unstable due to hydrolysis of nitrosonium cations. Binding of hydroxyl ions with them during this process induces conversion of nitrosonium cations into nitrite anions at neutral pH values, and therefore conversion of DNICs in the corresponding high-spin MNICs. In the presence of thiol-containing ligands in DNICs, which are characterized by high π-donor activity, electron density transferred from sulfur atoms to iron-dinitrosyl groups neutralizes the positive charge on nitrosonium cations, which prevents their hydrolysis, ensuring relatively a high stability of corresponding DNICs.

The presence of neutral NO molecules and nitrosonium cations in DNICs with thiol-containing ligands is detected based on their ability to be donors of these agents responsible for the corresponding biological activity of DNICs—a positive and regulatory effect due to NO molecules released from DNICs, or the negative cytotoxic effect related to nitrosonium cations released from DNICs.

How are the abovementioned components released from DNICs with thiol-containing ligands in living organisms? Is this related to direct contact of NO and NO^+^ targets (heme-containing and thiol-containing proteins respectively) with DNICs, or their contact after destruction of these complexes with relevant agents? To answer these questions, we need further research. Nevertheless, the research which we have conducted suggest that a shift in intracellular redox processes to oxidation that causes a decrease in thiol content can promote a release of both NO molecules and nitrosonium cations from DNICs. And vice versa, an increase in thiol content in case of activation of reduction processes can reduce the number of nitrosonium cations released from DNICs due to reduction of the latter to neutral NO molecules.

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
