# Peer review of "Physico-Chemistry of Dinitrosyl Iron Complexes as a Determinant of Their Biological Activity†"

_ijms, 2021, doi:10.3390/ijms221910356_

Round 1

Reviewer 1 Report

This manuscript reviews the literature of dinitrosyl iron complexes. In general, I consider the work is useful trying to show an actual topic in the field of medicinal chemistry and therefore will be of interest to the readers of International Journal of Molecular Sciences. Although the author has conducted a thorough literature review, undertaken a rigorous piece of data collection and analyzed the information accurately, the manuscript needs some attention prior to acceptance as is detailed below.

  1. There is a typo mistake: comlexes instead of complexes.
  2. Abstract. The specific acronyms used for mononuclear and binuclear DNICs should not be used before the general acronym, DNICs, be defined in the text.
  3. Introduction, lane 42. There is a typo mistake: DNIC instead of DNICs
  4. Introduction, lanes 43-45. I am not able to understand the sentence:

This novel compound…..¿are the authors referring to DNICs or to NO? Why novel?

  1. Introduction, lane 47. The oxide is not an element
  2. Page 6, Fig. 5. Since only panels B and D are mentioned in the text, panels A and C should be removed form the figure.
  3. Page 8, lane 327. There is a unknown symbol.
  4. Page 8, formula 3. Something is missing between lane 322 and 333.
  5. Page 9, schemes 1 and 2. Why are they schemes? Which is the difference with the former formulae?
  6. Figures 11, 12, 13, 15,17,18. The title of the y-axis is missing

Author Response

Response to reviewer comments.

Reviewer 1.

Dear Sir/Madam,

Thank you very much for your careful consideration of the MS and many useful comments. My response for your remarks is the following:

Remarks 1-3. Now I used acronyms M- and B-DNICs in all text of MS.

Remarks 4-5 (Lanes 43-47) I corrected the sentences here  as the following: . “These compounds  can serve  the basis for designing of medicines with  a wide range therapeutic action relying on the fact that nitrogen monoxide (or nitric oxide, NO) is one of the main components of DNICs [1-22]. This is a remarkably simple compound containing two elements that form part and parcel of Earth’s atmosphere: nitrogen and oxygen” and added them to the text in red fonts

Remark 6. I mentioned panels A and C in the text of revised MS (red fonts)

Remarks 7-9/ Now instead of Formulas 1-4 I used Equations 1-4. (red fonts)

Remark 10. I corrected the figures 11,12,13,15 and 17 in accordance with this remark.

 Besides all these corrections of the text of revised MS  I added to it a few sentences (section 2 and 8) by red fonts.

 I would  like to attract your attention (as well as attention of technical editor) to the following. Equation 3, Schemes 1, 2 and 7 are printed non-correctly in the text of proofs. Correct presentation of  Equation 3 and Schemes 1, 2 and 7 are shown in the revised MS by red fonts.

Reviewer 2 Report

This review by A.F.Vanin deals with a large series of great contributions of his own research from the past decades and is discussed in context with literature from other groups. It combines the history of the research on this field of chemistry (e.g., from the 1960s), the step-wise development of an understanding of mechanisms and in vitro and in vivo processes these compounds can undergo and modern investigations which aim at medical applications of this highly interesting and promising kind of compounds. First of all, I am a chemist with focus on coordination chemistry and molecular structures, I am not an expert with respect to some other fields which also play important roles in this review. Nonetheless, I found this manuscript a highly interesting read, written in a reader-friendly manner. In my opinion, it provides very good insights into the current knowledge of the chemistry of this class of compounds. Hence, I am happy with the structure of this review.

As to language, the text requires some minor polishing, which could easily be done at the editorial stage. The author, however, should carefully check the manuscript for remaining Cyrillic letters (e.g., in Figure caption 1, line 100, the Cyrillic “g” after 1964; or in the manuscript text line 114 the Cyrillic “I” between the g components. Also, some other minor errors (chemical things) should be tackled by the author rather than by the editorial or typesetting office, e.g., line 254 after “thiol-containing” the word “ligands” seems to be missing; sub-title 4.2. an “r” is missing in “nitrosonium”; in Scheme 10 “mononotrosyl” should read “mononitrosyl” and “dithocarmamate” should read “dithiocarmamate”.

As to the scientific contents, I found it puzzling in which way the iron electron configuration is presented. Line 297: “Fe2+ ion with six electrons on its 3d orbital,”: Well, I am basically happy with this expression (however, I would write “on its 3d level”, or maybe “in its 3d orbitals”) as I am used to seeing Fe2+ as a d6 system. Following the next lines (where one-electron oxidation or reduction is mentioned), I would expect d5 and d7 systems to be the new configurations derived therefrom. However, d7 or d9 are mentioned. Also, the description {Fe(NO)2}9 was, in my opinion, unusual and would require some explanation. Does the superscript “9” in {Fe(NO)2}9 refer to the non-bonding electrons in total (e.g., reduction of iron d6 to iron d7, combined with two molecules of NO, each of which bears a radical electron, thus amounting to 9 non-bonding electrons in the in {Fe(NO)2} moiety? If that´s the case, the expression in {Fe(NO)2}9 should not be used as a substitute for d9. Computational analyses of suitable model compounds, which would adhere to the expression in {Fe(NO)2}9, might still reveal that their 3d-orbital population is closer to 7 rather than 9.

Line 895: The resonance structure to NO+-RS- is NO-RS (not NO-RS-radical).

Lines 907/908: “…characterized by pink color with the absorption band at 334 nm.” Is misleading, it sounds like the pink color would be associated with this absorption band. Better write “…indicated visually by the appearance of a pink color and in UV/Vis spectra by an absorption band at 334 nm.”

Line 984: In Figure caption 22 a time delay of 40 ms is mentioned, but earlier (line 976) a delay of 20 ms is mentioned. Please check.

Line 997: A drop of 20 μL is mentioned, but earlier (line 974) a drop of 0.01 mL is mentioned. Please check.

Figures 25 and 27: In case of the bars of combined contributions (bar 3 in Fig. 25, bars 4 and 5 in Fig. 27) I would expect error bars which are noticeably wider than those of the individual contributions, however it looks like the size of the error bars has not changed. Please check.

Last but not least, as a coordination chemist I found it particularly interesting that the properties observed with these compounds are nicely in accord with square-planar coordination sphere of Fe(d7), and of course the strong ligands (NO)+ should support this coordination geometry. Then, I was surprised that all crystallographically confirmed examples of (RS)2Fe(NO)2 motifs showed essentially tetrahedral iron coordination spheres. I can only guess that additional ligands (maybe a nitrogen donor site within a protein matrix) might support the transition toward square-planar arrangement of (RS)2Fe(NO)2 (thus furnishing a square pyramid with a fifth ligand in apical position), and the single electron in dz2 might become more stabilized by interaction with a weak Lewis acceptor moiety (e.g., if it established a contact to a rather acidic proton). Hence, I can understand that in protein matrix this arrangement could be square-planar even though in crystals of model compounds it is tetrahedral. However, it is hard to understand as to why there are only tetrahedral examples found in crystals, but even in solution (where high mobility is possible and configurational exchange would contribute to entropy) the complexes should quantitatively switch to square-planar. With respect to this interesting question of complex structure, some lines might be added to the manuscript. Is there any literature which addressed the energetic difference between tetrahedral and planar isomers of such compounds (computational studies maybe)? Is there any computational study available which analyzed the 2.03 EPR signal with respect to the geometry of the complex (perhaps square-based pyramidal is more likely than square-planar without any additional weak ligand)? Even if the author, who is expert on this field of chemistry, had not come across such literature, it should at least be mentioned that those kinds of study have not been available in literature. This would help the reader understand as to why this question is not being discussed in more detail in this review. 

Author Response

Reviewer 2.

Dear Sir/Madam,

Thank you very much for your careful consideration of the MS and many useful comments. My response for your remarks is the following:

I made minor polishing of the text in accordance with your remarks mentioned in second section

As regarding the scientific contents which you mentioned in third section I corrected them keeping in mind the followings.  In accordance with Enemark-Feltham representation of iron-dinitrosyl group in DNIC – {Fe(NO)2}n, n is sum of number of electrons on iron 3d orbitals and 2 unpaired electrons on p-orbitals of NO molecules. For example, when Fe2+ ion combines with two NO molecules, n=8. It means that 8 electrons localize on MO combined from iron 3d orbitals and two p-orbitals of NO ligands. At two {Fe(NO)2}8 group disproportionation reaction (mutual one electron oxidation-reduction), these groups are transformed respectively into  {Fe(NO)2}7 and {Fe(NO)2}9 groups,  formally corresponding to d7 and d9 electron iron configurations ( if propose that all non-bonding electrons are localize on iron 3d orbitals in these groups). Naturally, that is very rough proposition, because these electrons are distributed on MO including both 3d orbitals of iron and p-orbitals of NO molecules.

Line 895. I propose that S-nitrosothiol molecules can be  represented with many resonance structures (depending on nucleophyle/electophyle nature of the molecules surrounding S-nitrosothiols in the solutions). Two such types of resonance structure – RS--NO+  and RS.-NO.

are used in Scheme 11.

Lines 907/908, 984, and 987: I corrected the text in accordance with your comments.

Figures 25 and 27. I beg your pardon for very rough mistakes on these figures, which were performed by my young co-author – student Darya Telegina (as regarding me I miss this mistakes) . Of course, while  combined contributions of columns 1 and 2 in Fig.25 and 1+2 and 1=3 in Fig.27 experimental mistakes should be summarize. I made this corrections.

As regarding proposed transformation of DNIC tetrahedral structure transformation into square-plane one at DNIC crystals dissolving which I suggest in my MS I plan to carry out detailed analysis of this problem in the future.